Journal of
open psychology data

# Curation of FOAMS: a Free Open-Access Misophonia Stimuli Database

DATA PAPER

**DEAN M. ORLOFF**

**DANIELLE BENESCH** (iD)

**HEATHER A. HANSEN** (iD)

*Author affiliations can be found in the back matter of this article

]u[ ubiquity press

## ABSTRACT

Misophonia is a disorder of decreased tolerance to certain "trigger" sounds (e.g., chewing, tapping, clicking). While misophonia research is scant in general, studies presenting sounds are especially rare and methodologically variable, likely due to the labor and time required to create stimuli. Thus, we introduce FOAMS: Free Open-Access Misophonia Stimuli, a sound bank publicly available on Zenodo, accompanied by pilot discomfort ratings for 32 of these sounds (4 exemplars of 8 classes). The FOAMS database aims to decrease the burden on researchers, facilitating reproducibility and the pursuit of nuanced research questions to better understand this perplexing disorder.

**CORRESPONDING AUTHOR:**
**Heather A. Hansen**

Department of Psychology, The Ohio State University, Columbus, OH, USA

hansen.508@osu.edu

**KEYWORDS:**
Misophonia; sound sensitivity; stimulus set; sound bank; database

**TO CITE THIS ARTICLE:**

# (1) BACKGROUND

Misophonia is a disorder of decreased tolerance to specific sounds or stimuli associated with those sounds (Swedo et al., 2022). Although often thought of as an aversion to oral/nasal sounds in particular (Jager et al., 2020; Kumar et al., 2021; Schröder et al., 2013), large-scale surveys and experimental investigations of misophonia have revealed a wide variety of reported triggers: chewing, sniffling, keyboard typing, rustling plastic/paper, cutlery noises, etc. (Cavanna & Seri, 2015; Hansen et al., 2021; Vitoratou et al., 2021). Misophonia is a prevalent disorder – population studies estimate 5-20% of the general population is affected (Vitoratou et al., 2023; Kılıç et al., 2021; Jakubovski et al., 2022) – and leads to significant impairment in daily life activities for sufferers (Rouw & Erfanian, 2017; Swedo et al., 2022). Consequently, studying the disorder and its effects has been a focus of recent research.

To experimentally study a disorder of sound processing, one logical approach is to present sounds to participants; however, only about a dozen studies so far have incorporated sound stimuli. We surmise this is true for a few reasons: First, except in rare instances of collaboration, each research team must start from scratch in compiling their own stimulus sets, which is both time and labor intensive. Additionally, perhaps due to the overwhelming range of sounds that misophonic individuals find bothersome, existing studies in the literature vary widely in which sound stimuli are used in their "triggering" condition. For example, some studies use primarily oral/nasal sounds, such as chewing or sniffling (Daniels et al., 2020; Edelstein et al., 2020; Kumar et al., 2017; Savard et al., 2022; Siepsiak et al., 2023), whereas others equally incorporate non-oral/nasal or nonhuman sounds as triggers (Enzler et al., 2021; Grossini et al., 2022; Hansen et al., 2021). While useful starting points, both approaches have drawbacks: The latter is more time-consuming and may include trigger sounds that not all misophonic individuals are personally averse to, but the former may not sufficiently capture the variation in misophonia and subsequently isolate individuals who experience triggers that are less common. Similarly, most experiments incorporating stimuli thus far have included single instances of each trigger sound (e.g., Daniels et al., 2020; Enzler et al., 2021; Grossini et al., 2022; Hansen et al., 2021; Heller & Smith, 2022; Kumar et al., 2017; Seaborne & Fiorella, 2018; Silva & Sanchez, 2018). While simpler methodologically, individuals with misophonia often report varied reactions to the same sound produced by different sources (e.g., more aversion to a loved one chewing than a stranger chewing (Edelstein et al., 2013, 2020)); thus, using a single exemplar introduces uncertainty regarding whether the particular stimulus chosen will feel bothersome to a participant with that trigger.

In addition to the differences in stimulus content, wide variability exists in other choices regarding how misophonia stimuli are presented. For instance, previous misophonia research has used sound stimuli that vary in duration from around 2 seconds (e.g., Enzler et al., 2021) to 30 seconds (e.g., Grossini et al., 2022) and in breadth from fewer than 5 trigger sounds (e.g., Heller & Smith, 2022) to more than 100 potential triggers (e.g., Hansen et al., 2021). Thus, it is unclear whether any resultant effects are attributable to misophonia or merely a byproduct of how long (or how many) stimuli were listened to. Similarly, low-level acoustic properties can also affect responses, as stimuli played at excessively high volume levels can be bothersome to most listeners (Kaernbach et al., 2011; Skagerstrand et al., 2017), regardless of their misophonia status. Additionally, stimuli played at low volume levels may also affect participants' responses if they are inaudible enough to be recognized, since the misophonic reaction has been shown to be dependent upon sound identification (Edelstein et al., 2020; Hansen et al., 2021; Heller & Smith, 2022; Savard et al., 2022). Thus, different recordings of the same trigger sound may evoke different experimental responses due to variations in acoustics and context. Without access to the original stimuli, replication of individual studies is challenging.

Taken together, the field would benefit greatly from some common ground on which to study misophonia. To fill in this gap, we present the FOAMS database—Free, Open-Access Misophonia Stimuli. The FOAMS database seeks to 1) increase the quantity of current, usable stimuli in the misophonia research field, 2) standardize the stimuli used in experimental research, and ultimately 3) establish a free and open-access platform to aid in reproducibility of existing and future studies. This paper first presents the development of the database, including how specific categories of trigger stimuli were collected and labeled. This process provides detailed information regarding the name, duration, salience, and category of each trigger sound. Additionally, this paper presents pilot aversiveness ratings from a subset of categories in the database as part of a larger cognitive experiment, demonstrating the flexibility and utility of the database for specific research questions. Importantly, while standardized aversiveness ratings (e.g., valence, arousal) for all the sounds would make FOAMS maximally useful, these ratings data and the FOAMS sound bank function as a proof of concept, exemplifying how open-access sound stimuli may be utilized in future experimental research of misophonia.

# (2) METHODS

## 2.1 MATERIALS
### 2.1.1 FOAMS creation
To create the initial release of the FOAMS database, sound search terms were compiled using previous

research and prioritized based on how well discomfort to the sound correlated with misophonia severity (Hansen et al., 2021). Specifically, we used the sound list provided by Figure S8-B of Hansen et al. (2021), which presents a Misophonia Sensitivity Index depicting the correlation between participant discomfort ratings for each sound and participant misophonia levels. Higher correlations indicate sounds that better discriminate individuals with misophonia from controls. Thus, sounds that were significantly correlated with misophonia severity after correcting for multiple comparisons were prioritized for the FOAMS sound bank. Although 37 sounds met this criterion and were considered, 10 unique low-level classes of sounds were chosen for the initial release of FOAMS given the effort required to manually annotate them (see below), as done by the UrbanSound8K database (see Salamon et al., 2014).

Sound stimuli were then compiled from existing files via freesound.org, with up to 1000 results for each search term. Stimuli were further prioritized if the following criteria were met: the sound lasted from four to 150 seconds, was released under a creative commons zero license, had a sampling frequency of at least 44100 Hz, and was the first submission from a unique Freesound user after sorting by priority. For sounds in which the label from Hansen et al. (2021) did not return five usable instances from freesound.org (e.g., fewer than five instances in search results, or at least five instances in search results but fewer than five that met pre-established criteria), the term was removed. Of the search results, each sound was further categorized following priority order: Each sound was listened to by a member of our research team, who determined if 1) either the desired sound (or another relevant sound) was indeed present in the audio file for at least four seconds, and 2) if the audio file contained interfering background noise. This process continued until five sounds from ten unique categories were found to have the desired search term present without background noise. The ten categories presented in the initial release of FOAMS include chewing gum, flipping newspaper, typing, basketball dribbling, knife cutting, human breathing, plastic crumpling, water drops, clearing throat, and swallowing.

Each sound event for the audio file was annotated using Audacity® (v3.1.3). The entire audio file was manually labeled, marking every occurrence of the desired primary sound as well as any additional secondary sounds, aiming to annotate the onset and offset of each sound event within 50 ms precision. Labels also denoted salience, with "C1_sound" and "C2_sound" indicating foreground or background sounds, respectively. These techniques were modeled after those used to construct the existing UrbanSound8K database (Salamon et al., 2014). Then, representative four-second instances of each sound were selected by a member of the research team. During this process, the labeled audio file was listened to again and all previously-labeled instances of the target sound were considered for the segment creation. Instances with minimal background noise and the most isolated target sound were selected, aiming for about four seconds long, with slight variation so as to not cut the sound off abruptly. A four second duration was chosen because of its use in the UrbanSound8K database, based on four seconds being sufficient for participants to identify the sound (Salamon et al., 2014, Chu et al., 2009); previous misophonia literature has shown the critical role of sound identification (Savard et al., 2022, Heller & Smith, 2022). One segment was chosen for each of the 50 stimulus files. The initial labels and final segmentation were both exported to TXT files from Audacity and are publicly available on Zenodo.

Finally, a taxonomy was created based on the search terms and updated throughout the annotation process. As with annotation, we modeled our taxonomy method on UrbanSound8K, given a misophonia taxonomy does not yet exist and UrbanSound8K's categorization of environmental sounds is similar in content to misophonia triggers. We started with the parent categories used in their taxonomy of urban sounds (e.g., "Human", "Nature", and "Mechanical") then modified the taxonomy using our search terms to adapt it for misophonia-relevant sound categories (e.g., adding "oral/nasal" as a subheading under "Human"). Each of the search terms we considered as categories in the initial release of FOAMS (see description above, Hansen et al., 2021) was added to the taxonomy under the appropriate subheading. The taxonomy was updated during the annotation process so that each label present in an audio file was included; for example, "exhaling" was not a prioritized search term, but was present in the audio files, so it was added to the taxonomy. Each label was categorized under at least one parent sound that described the type of sound present in the label; researcher discretion was used to determine relevant parent categories. While this taxonomy is far from exhaustive and may be rearranged as more sounds are added, we find this structure useful in further categorizing sounds that have relatively broad labels. For example, a parent term called "oral/nasal" might be relevant to researchers who are studying the effects of oral/nasal sounds more generally, but the parent term could be further subdivided into more specific labels like "lip smacking," "chewing," or "swallowing" for researchers who have a narrower focus. In this sense, the taxonomy and label files provided in FOAMS are useful for a plethora of research questions. We have supplied the taxonomy in JSON format on GitHub to facilitate use, extension, and reorganization by future research.

### 2.1.2 Pilot stimuli

Pilot discomfort ratings were derived from a larger experiment studying the cognitive and social effects of misophonia via a face memory task (see Hansen et al., n.d.).

For the purposes of that experiment, eight human-produced classes from FOAMS were used (breathing, chewing gum, clearing throat, swallowing, knife cutting, basketball dribbling, flipping newspaper, and typing). To approximately equate sound durations between classes, four instances from each class were chosen by removing the shortest or longest instance from the available five. To control for sound exposure, these 32 sound stimuli from FOAMS were supplemented with four instances of pink noise and four trials with no sound, resulting in 10 total classes with four instances each.

The FOAMS stimuli do not have sound level normalized, both to give researchers flexibility and maintain variability when sound level is a factor of interest. However, for this cognitive experiment, sound level variability was not a factor of interest. As such, sound levels were normalized using Adobe Audition (v.14.4) by matching Total RMS to the first chewing gum file; chewing gum was chosen for its quieter starting volume and role as a classic misophonia trigger. Total RMS was -50.03dB for each sound used in the pilot.

## 2.2 SAMPLE

21 participants (Mean Age = 18.5; 11 female, 8 male) were recruited for the pilot. Participants were undergraduate students who were enrolled in an Introduction to Psychology course at The Ohio State University and received course credit for their participation.

Participants were assessed for misophonia using the Duke Misophonia Questionnaire (DMQ; Rosenthal et al., 2021) and Selective Sound Sensitivity Syndrome Scale (S5; Vitoratou et al., 2021), of which misophonia is suggested to be present above scores of 87 out of 250. Our 21 participants had a mean S5 score of 44.8 (range: 0-134); four participants scored above the 87 criterion, matching prevalence estimates of misophonia in undergraduate samples (Wu et al., 2014; Zhou et al., 2017) and the general population (Vitoratou et al., 2023).

## 2.3 STUDY DESIGN

The experiment was run in a dimly lit, sound-attenuated testing room using a Mac Mini computer with a 24-in. LCD monitor. Stimuli were presented using Python 3.8 and PsychoPy (v2021.2.3). Before beginning the experiment, participants were informed that they would be presented faces and asked to make judgments about the faces. Participants were made aware that sounds would play concurrently with the faces, and that some sounds may feel unpleasant to them.

The experiment was broken down into two parts: Phase 1 (Learning) and Phase 2 (Memory). In Phase 1, participants were shown 40 faces one at a time while completing an incidental encoding task. During presentation of the face, a stimulus from one of 10 sound classes played aloud through speakers. Afterwards,

participants were shown a response screen on which they were given two additional tasks: 1) judge the identity of the sound they just heard by choosing one of the 10 available class names, and 2) rate their discomfort during the sound on a scale from 0 (no discomfort) to 5 (max discomfort). After clicking responses to both questions, a "Continue" button appeared, after which participants started the next trial. Participants were given two practice trials (one male face, one female face) accompanied by pink noise (labeled "white noise" on the screen for familiarity), then completed 80 experimental trials split into 4 blocks, between which they were offered short breaks. In Phase 2, participants made trait and memory judgments about the faces from Phase 1; results from this phase are outside the scope of the present paper.

## 2.4 ETHICAL ISSUES

All research was approved by the Institutional Review Board at The Ohio State University. Participants provided informed written consent prior to data collection and were assigned an anonymous ID number for data storage.

## 2.5 EXISTING USE OF DATA

The FOAMS database was used in a dissertation experiment conducted by a member of the research team, currently under review for publication (Hansen et al., n.d.).

# (3) DATASET DESCRIPTION AND ACCESS

## 3.1 REPOSITORY LOCATIONS

FOAMS DOI: 10.5281/zenodo.8170225
Pilot DOI: 10.5281/zenodo.8170180

## 3.2 FILE NAMES
### 3.2.1 FOAMS

- FOAMS_documentation.pdf: details of audio labeling, segmentation, and taxonomy creation
- FOAMS_processed_audio.zip: all labeled stimuli available in the database, in WAV format
- FOAMS_processed_audio_flac.zip: all labeled stimuli available in the database, in FLAC format
- segmentation_info.csv: details of stimulus segments

### 3.2.2 Pilot

- Sub01.csv – Sub21.csv: raw experimental output of discomfort ratings and sound identifications for all 21 participants
- MisoAssessments.csv: DMQ and S5 assessment scores for all 21 participants
- Stim_reference_table.csv: reference table of the sound stimuli with their corresponding FOAMS IDs
- FOAMS_analysis.m: analysis script for compiling raw data and generating a summary table

- discomfort_summary.csv: summary table of discomfort ratings for the 32 FOAMs sounds used in the pilot
- README.txt: explanation of the raw experimental output files
- Pilot_sound_stimuli.zip: all 33 sound stimuli used (32 from FOAMS + pink noise), in WAV format
- Pilot_sound_stimuli_flac.zip: all 33 sound stimuli used (32 from FOAMS + pink noise), in FLAC format

### 3.3 DATA TYPE
Primary data, processed data

### 3.4 FORMAT NAMES
Sound files are available in both WAV and FLAC audio formats. Pilot data is available in CSV format. Analysis scripts of the pilot data are available for use in Matlab (version R2021a).

### 3.5 LANGUAGE
American English

### 3.6 LICENSE
Creative Commons Attribution 4.0 International Public License

### 3.7 PUBLICATION DATE
September 25, 2022

## (4) REUSE POTENTIAL

The FOAMS database and pilot discomfort ratings provide numerous interdisciplinary benefits. Firstly, since the FOAMS database has multiple exemplars of each sound with varied acoustic properties, this database can enable more nuanced research questions. For example, auditory researchers may use the differential discomfort ratings assigned to the four piloted chewing sounds to explore which acoustic properties (e.g., frequency, intensity) best explain why some instances of the trigger sound are more aversive than others. Furthermore, the FOAMS database's diverse collection of sound exemplars with varying acoustic properties presents an opportunity for machine learning research. With its diverse collection of sound exemplars, researchers could leverage this sound bank to develop robust machine learning models for automatic detection of misophonic triggers, opening avenues for personalized interventions and advancements in managing misophonia (Benesch et al., 2021). By modeling the FOAMS format to match that of UrbanSound8K, a popular dataset used in sound event classification research, we hope to encourage the use of FOAMS in the machine learning community.

More generally, an open-access database will bridge gaps in misophonia literature and make results more interpretable. For instance, if neuropsychological studies from different research groups present these sounds to participants and observe conflicting results, researchers can be more confident the disparate findings are not merely confounded by the particular stimuli each group presented. Additionally, given the individual differences in misophonic experiences, researchers could benefit from individually tailoring their experiments to each individual's trigger sounds, an ideal put forth by Schröder et al. (2019). Importantly, all files used to create the final processed dataset have been made publicly available, including the sound search results, the original audio files, the annotation files, and the taxonomy, which provides transparency and facilitates replication. This information offers much potential for expansion or modification of the FOAMS database if researchers need to include more sounds or tailor the preprocessing to their own specifications. This is relevant given that the initial release of the FOAMS database contains 10 sound categories, and misophonic individuals report a plethora of triggers; as such, not all trigger sounds are presently represented in the database, and further expansion would make it maximally useful.

Aside from research purposes, sounds from this database can be used in diagnosis, therapy, and a broadened awareness of misophonia in the medical community. Enzler et al. (2021) demonstrated an ability to assess misophonia by analyzing ratings of pre-selected sounds (see also Hansen et al., 2021). With a larger and more diverse sound bank, the success in capturing different variations of misophonia improves. Moreover, although about 20% of undergraduate samples (Wu et al., 2014; Zhou et al., 2017) and the general population (Vitoratou et al., 2023) experience misophonia, not all treatment providers are comfortable with the term; in a study of audiologists in India, only about 15% of them reported confidence in handling the condition (Aryal & Prabhu, 2023). Often a multidisciplinary treatment team is preferred (Aryal & Prabhu, 2023), with psychologists using therapies that may incorporate stimulus presentation (see Mattson et al., 2023 for a review of treatments). Freely accessible sound stimuli can thus be incorporated into training seminars or individual therapy plans to familiarize treatment providers with the disorder and improve treatment outcomes.

The FOAMS database is a compilation of existing sound files and is therefore intrinsically limited in its scope. That is, all categorized sounds come from existing, user-uploaded audio files on freesound.org; no sounds were recorded by the research team. This reliance on previously existing sound files presented logistical challenges when analyzing certain sound

categories, since not all desired categories (e.g., "sipping hot liquid") had search results on freesound.org, or search results contained multiple categories besides the desired sound (e.g., "slurping" containing lip smacks and swallowing sounds, or "lip smacks" containing audio indistinguishable from chewing gum). Further, acoustic properties (e.g., due to the recording device, background noise) could not be controlled. The reliance on freesound.org also necessitated the use of researcher discretion when annotating sounds to verify that the content matched the user-uploaded description and to choose representative segments of each audio clip. Finally, while offering five exemplars of each sound category is more ecologically valid than presenting just one sound, doing so cannot fully account for the idiosyncrasies of the misophonic experience, especially for sufferers who are mainly bothered by sounds from select individuals (e.g., family/friends, Edelstein et al., 2013).

Despite these limitations, this intrinsic structure of the FOAMS database fosters both flexibility and reproducibility in research; because FOAMS relies on existing sound databases, the potential for expansion remains feasible via the aforementioned methods. Further, the acoustic variations in sounds—though at first apparently confounding—enables researchers to examine more specific issues and is not necessarily a limitation of the FOAMS database. For example, a study using only "swallowing" sounds could examine what specific characteristics of each swallowing sound make it triggering; is it variation in background noise? Does the sound quality affect trigger response? The reuse potential is wide, and more open-access resources like the FOAMS database will benefit the misophonia field as a whole. This proof of concept lays the framework for such broad, reproducible, and collaborative future efforts in misophonia research.

## ACKNOWLEDGEMENTS

We thank soQuiet for financially supporting this research project, as well as the many users on freesound.org who uploaded sounds we could use in this database. We would also like to thank our collaborators on the project, Marie-Anick Savard, Emily Coffey, and Mickael Deroche, for their helpful suggestions. Lastly, we thank Andrew Leber and Zeynep Saygin for supervising data collection of the pilot.

## FUNDING INFORMATION

Funding for the creation of this sound bank was provided by a 2022 soQuiet Misophonia Student Research Grant awarded to DB and HAH.

## COMPETING INTERESTS

The authors have no competing interests to declare.

## AUTHOR CONTRIBUTIONS

DO curated the FOAMS database (including sound search, sound labeling, and sound segmentation) and drafted the manuscript. DB developed software to automatically segment and process the FOAMS stimuli. HAH collected pilot discomfort ratings. Both DB and HAH conceptualized and supervised the project and edited the manuscript.

## AUTHOR AFFILIATIONS

**Dean M. Orloff**
Department of Psychology, The Ohio State University, Columbus, OH, USA

**Danielle Benesch** orcid.org/0000-0002-2002-2325
ÉTS-EERS Industrial Research Chair in In-Ear Technologies, Montreal, QC, CA

**Heather A. Hansen** orcid.org/0000-0002-8917-2516
Department of Psychology, The Ohio State University, Columbus, OH, USA

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

## PEER REVIEW COMMENTS

*Journal of Open Psychology Data* has blind peer review, which is unblinded upon article acceptance. The editorial history of this article can be downloaded here:

- **PR File 1.** Peer Review History. DOI: https://doi.org/10.5334/jopd.94.pr1

**TO CITE THIS ARTICLE:**

**Submitted:** 24 May 2023   **Accepted:** 26 July 2023   **Published:** 29 August 2023

