## [Peer Review History. · Journal of Open Psychology Data]

Peer Review Comments for "Curation of FOAMS: a Free Open-Access Misophonia Stimuli Database"

Dear Dean M. Orloff, Danielle Benesch, Dr. Heather A. Hansen,

After review, we have reached a decision regarding your submission to Journal of Open Psychology Data, "Curation of FOAMS: a Free Open-Access Misophonia Stimuli Database". Our decision is to request revisions of the manuscript prior to acceptance for publication.

The full review information is included at the bottom of this email. Please note that there may also be a copy of the manuscript file with reviewer comments available once you have accessed the submission account. We ask you to please consider the following issues and revise the file accordingly:

In sum, we've had some excellent reviewers from this field and they have a number of suggestions which I consider worth reflecting upon. For example, reviewer C encourages you to check that the format type of the stimuli are the most accessible, and reviewer A has a number of field-specific suggestions for how to refine the manuscript. As such, please go through this feedback systematically and respond in a 'response to reviewer' document, making all changes using 'tracked changes' or coloured text so we can ensure the next part of the process is swift for you.

Instructions for how to resubmit your article online are pasted below. Please ensure that your revised files adhere to our author guidelines, and that the files are fully proofed prior to upload. Please also include a revised version of your article with 'tracked changes', adding comments where appropriate, to indicate the revisions made, in addition to a brief document outlining how you have responded to the reviewers' requests.

If you have trouble processing the revisions, our Help Center (<https://help.u-community.io>) or downloadable PDF (<https://bit.ly/Author-Guide-OJS-3>) may be able to help. If not, please get in touch and we'll be happy to help.

Please also ensure that all copyright permissions have been attained for any figures/tables you have included.

Please could you have the revisions submitted with three weeks. If you cannot make this deadline, please let us know as early as possible.

Kind regards,

Dr Thomas Rhys Evans

Reviewer A:
Recommendation: Revisions Required

Dear authors of the article “ Curation of FOAMS: a Free Open-Access Misophonia Stimuli Database”,

First, I would like to thank you for your great work and effort you put in developing the FOAMS. I think the paper tackles a huge gap in misophonia research and is a significant contribution to the field. With your work you have laid the foundation for a unified stimuli usage and development. I seriously hope that the data base is used and steadily extended by peer researchers in the field.
Please find my review below.

Overall evaluation:

The article provides a comprehensive and exhaustive description of the newly created FOAMS database. The methods for the FOAMS creation are well described in large parts of the manuscript. However, there are some important details missing that I find necessary to report. This mainly regards the initial release of the FOAMS, the sound class determination and the description of the taxonomy. Before publishing the article, I deem it necessary to add these details. An outstanding strength of the article is its description of the potential reuse and extension of the FOAMS database which is extremely relevant for misophonia research. Given the quality and relevance of the FOAMS database, I strongly recommend its publication, provided that the rather minor issues have been addresses.

Detailed evaluation by paragraph and sub-paragraph:

Paragraph (1) Background

I'd rather not describe misophonia as a “newly defined” disorder. Perhaps it's better to remove “newly defined” as it's not adding anything substantial.

You should cite more recent findings on misophonia triggers (e.g., your own article: Hansen et al., 2021; and Vitoratou et al. (2021); IRT investigation of triggers). Also, I do not see the reason to cite case studies for a general property of misophonia, because there is valid evidence that non-oral/nasal sounds are triggers for misophonia. In my opinion it is more reliable to cite large-scale studies and methodologically strong investigations of misophonic trigger sounds here. In the next sentence, you say that misophonia is highly prevalent in the general population, but cite two articles that investigated a non-representative student sample as opposed to the general population. At this point, I'd recommend citing Jakubovski et al. (2022) & Pfeiffer et al. (2023) (German representative prevalence estimates) and Kılıç et al. (2021) (Turkish prevalence estimates). Since prevalence estimates in the general population vary from 5 to 20% (also considering that there is no standard in misophonia diagnostics which might explain this variation), I'd not say “highly” prevalent. You could still say it's a prevalent disorder with estimates ranging from 5 to 20 %, being a bit more modest here.

I really like the argumentation line you are building up in the subsequent paragraph. You elaborated all important shortcomings of previous stimulus material. However, at the end of the paragraph you are saying that the pilot aversiveness ratings serve the

purpose of exemplifying the utilization of FOAMS, which is absolutely true, but I'd definitely add that for the standardization process sound aversiveness ratings are essential. In line with other standardized sound basis (e.g., IADS), you should highlight that sound ratings are a crucial standardization property which is pending for the FOAMS.

In that same sub-paragraph you are explaining that individuals with misophonia often report varied reactions to the same sound produced by different sources and that using a single exemplar of a stimulus entails uncertainty about whether the sound is perceived as bothersome. Your solution for that is providing multiple instances of a sound, which is great for many reasons, but it only slightly reduces the uncertainty introduced by the idiosyncrasy of individuals (especially the differences due to the relationship with the person producing the sound). This is not a problem of your stimuli, but rather a general problem in misophonia research, but I'd highlight at some point of the article that four different instances is not a remedy for the problem of idiosyncrasy of misophonic triggers.

Paragraph (2) Methods

Section 2.1.1. FOAMS creation

Please describe the criteria you chose for the initial release of the FOAMS database more detailed. Did you use a specific correlation cutoff and why?

How did you determine the ten sound classes? Was there any rationale or did you rely on specific misophonia articles? I'm wondering why the categories are varying in their broadness as some are extremely specific (e.g., basketball dribbling) whereas others are very broad (e.g., typing). I think it can easily be explained by the rationale I was asking for. Also, there are some important trigger sound categories missing (e.g., coughing, sniffing, clearing throat). I'm not saying they should definitely be in the database right now, but I'd need some more explanation behind the choice of trigger categories.

I could not find a paragraph on the volume of the stimuli, although you have greatly outlined the importance of volume for misophonic stimuli in the preceding paragraph of your manuscript. Please add the volume level, if there is a standardized volume level, and why you chose that level.

The annotation of the stimuli is very comprehensive and helpful. Thank you for the great work here!

I like the idea of a taxonomy, but I have some issues with it. I can't find an argument on why you chose the UrbanSound8K as a blueprint for your taxonomy. For example, why is this a good starting point for a misophonia-specific taxonomy of triggers? To understand the subjective choice of semantic classes, this paragraph and also the paragraph in your documentation would need some further details. These sections would greatly benefit from clarifying from which specific (misophonia) articles you have got your inspiration or on which you based your taxonomical changes. In my opinion you should give more insight into the development process.

Section 2.2 Sample

Again, Wu et al. (2014) and Zhou et al. (2017) are no prevalence estimates. Do not cite them as prevalence estimates. You can still make the point that your sample matches the prevalence estimate of Vitoratou et al. (2023).

Section 2.3 Study design

Please indicate the PsychoPy version you used.

I really like that you give a comprehensive example of the application of the FOAMS. Also, I really appreciate that you provide a summary on the discomfort ratings as researchers might want to choose sounds based on their discomfort level and as I said before this is an essential standardization property as well. So, as far as you have discomfort rating data on all FOAMS stimuli, please publish that as well.

Paragraph (4) Reuse potential

The only issue that needs to be mentioned here is that the statement about misophonia prevalence should be corrected in this paragraph, too. The paragraph provides concrete and useful suggestions for reuse of the data.

Deposited data

The deposited data meets all JOPD criteria.

With best regards,
Nico Remmert

Reviewer B:

Recommendation: Accept Submission

I thoroughly appreciated your work, and I consider it a significant contribution to the field. As you have highlighted, the existence of such a database is crucial for advancing experimental studies. I downloaded and listened to the sounds, and as someone who experiences misophonia, I can confirm that they are indeed as annoying as they should be! As a researcher on the field I can only thank you for this valuable contribution.

Reviewer C:
Recommendation: Accept Submission

This paper sought to address some of the methodological limitations of using and/or wanting to use sound stimuli to examine misophonia (for example, the time-consuming properties of researchers producing their own stimuli, the variability between the methodology of previous studies, etc). A proposed solution to assist in addressing these methodological struggles is FOAMS: Free Open-Access Misophonia Stimuli, a sound bank publicly accessible on Zenodo. Overall, this paper appears extremely relevant (with the researchers also highlighting its relevance concisely within the 'background'), and is a well-produced, well-structured explanation of how the FOAMS database was constructed.

How the paper meets the JOPD peer-review criteria
(<https://openpsychologydata.metajnl.com/about/editorialpolicies/>):

1a. The methods section is of a high quality enough so that within reason it could be reproduced by future researchers attempting to add and/or produce a similar database.

1b. The dataset is accurately described and meets the FAIR principles.

1c. The 'reuse potential section' is excellently written, illustrating a plethora of uses for FOAMS which would expand the field of misophonia research (as well as within therapeutic settings - working towards increasing the confidence and awareness of misophonia in the medical community).

2a. The dataset 'repository' is suitable for the subject and employs a sustainable model. I was able to access the data easily and adequate detail was provided to understand the file label system.

2b. The data is deposited under a CCo which permits unrestricted access.

2c. The data is in a .wav (propriety format). Personally, I am unsure as to if there is a non-propriety format that the researchers could provide (as I am not an expert in this), this is only required to do so if possible.

2d. The deposited data is labelled in a way that is readable to 3rd parties and with the documents the researchers provide it is a fairly easy process to understand.

Thus, again abiding to the FAIR principles.

2e. All necessary documents are provided to make the data accessible.

2f. This research abides to the American Psychological Association (APA) Ethical Principles of Psychologists and Code of Conduct

(<http://www.apa.org/ethics/code/index.aspx>), successfully anonymising the participant's data (using anonymous IDs).

Overall, as you can see, I feel the paper does meet the JOPD peer review guidelines and should be accepted for submission. This is a wonderful addition to the misophonia literature and hopefully with its implementation we will see a more standardised, specific, and replicable set of methodologies for misophonia research.